# Overview of Standards Related to the Occupational Risk and Safety of Nanotechnologies

**Delfina Ramos** [1,2,*] and **Luis Almeida** [3]

1    Institute of Science and Innovation in Mechanical and Industrial Engineering (INEGI),
     Campus da FEUP Rua Dr. Roberto Frias, 4200-465 Porto, Portugal
2    Algoritmi Centre, School of Engineering, University of Minho, 4800-058 Guimarães, Portugal
3    Centre for Textile Science and Technology, School of Engineering, University of Minho,
     4800-058 Guimarães, Portugal; lalmeida@det.uminho.pt
*    Correspondence: dgr@isep.ipp.pt

**Abstract:** Nanomaterials offer new technical and commercial opportunities but, due to their low particle size, raise occupational health and safety concerns and may also pose risks to the consumers and the environment. In the last 15 years, many standards have been developed in the area of nanotechnologies, taking into account, namely, occupational risk and safety. This paper presents an overview of the standards in this area, with special emphasis at the ISO level, but also at European level, where standards are considered as an important support for legislation. A brief presentation of five relevant ISO standards is included. Relevant European Standards are also mentioned. The control banding approach for occupational risk management applied to engineered nanomaterials, according to ISO/TS 12901-2:2014, is presented. Standards are essential for society and should, in fact, be considered an important tool for companies to support sustainable products and process innovation.

**Keywords:** standards; nanotechnology; risk; health and safety

## 1. Introduction

Nanotechnology has been promoted as the "next big thing" that will transform everyday life through the creation of numerous new products and enhanced materials for improved quality of life [1].

Although there are still few regulations regarding consumer products of nanomaterials, international organizations and the developed countries are trying to design guidelines and standards for toxicity evaluation and regulation plans, taking into account the nano-safety of humans and the environment. There is uncertainty on how nano-regulations may affect future funding, research, and development in the nano field, and this may lead to a delay in the commercialization of nanoproducts [2]. The regulations need the support of standards, namely in the areas of definitions, test methods, and specifications. There has been a strong increase in the development of standards in the area of nanotechnologies, especially since 2005.

According to National Nanotechnology Initiative (NNI), around the world, there are numerous standard-setting groups that are involved in developing nanotechnology standards. Some of the leading standard-setting organizations and their relevant nanotechnology committees are the International Standardization Organization (ISO) Technical Committee (TC) 229 on Nanotechnologies, ASTM International's Committee E56 (Nanotechnology) (formerly known as the American Society for Testing and Materials), the International Electrotechnical Commission Technical Committee 113 (Nanotechnology Standardization for Electrical and Electronics Products and Systems), and the Institute of Electrical and Electronics Engineers' Nanotechnology Council. These groups develop voluntary standards. Standards that are the best formulated, with the strongest basis in

science, are most likely to be adopted by the global community. The U.S. also holds leadership of the ISO TC 229's Working Group 3: Health, Safety, and Environmental Aspects of Nanotechnologies, with a representative from The National Institute for Occupational Safety and Health (NIOSH) [3].

At the European level, the Technical Committee CEN/TC352—Nanotechnologies is linked to its international counterpart, ISO/TC 229—Nanotechnologies, and to the Committees in charge of nanotechnologies within the EU Member States' National Standardization Bodies. CEN/TC352 has four working groups, one of which, WG3, deals with health, safety, and environmental aspects.

Occupational Health and Safety aspects are considered as very important for the European Commission. Nanomaterials offer new technical and commercial opportunities but, due to their low particle size, raise occupational health and safety concerns and may also pose risks to the consumers and the environment [4–6]. The European definition of nanomaterials emphasizes the concerns about health and safety and is an important basis for all the legal requirements related to nanomaterials.

In this study, an overview of the standardization related to the Occupational Risk and Safety of Nanotechnology is presented.

The ISO standard describing the use of the control banding approach for occupational risk management applied to engineered nanomaterials has been used in a case study in a textile finishing company.

### 1.1. Overview of Nanotechnology in the World

This section is based on the information disclosed by the company StatNano, namely the "Standard Database" on standards and publications related to nanotechnology, with the status, level, year, organization, country, and classification.

According to StatNano, 1422 nanotechnology standards have been published by consensus and approved by 42 recognized bodies since 1992 [7]. Table 1 shows a conspicuous drop since 2019, when 368 standards were published, most of which were adopted.

**Table 1.** The number of published nanotechnology standards between 2018 and 2022 [7,8].

| Year | Number of Published Standards |
|------|-------------------------------|
| 2018 | 237 |
| 2019 | 368 |
| 2020 | 339 |
| 2021 | 335 |

StatNano [8] recently presented a survey of nanotechnology publications in 2021. Table 2 presents the total number of nano-articles published in 2021 by the first 10 countries, as well as the share of the nanoparticle articles in relation to the total number of articles.

**Table 2.** Number of nanotechnology publications in 2021 per Country [9].

| Ranking | Country | Total Number | Share of Nano-Articles to Total (%) |
|---------|---------|--------------|-------------------------------------|
| 1 | China | 85,758 | 14.1 |
| 2 | USA | 23,225 | 4.5 |
| 3 | India | 19,041 | 13.9 |
| 4 | Iran | 11,196 | 18.7 |
| 5 | South Korea | 10,355 | 12.9 |
| 6 | Germany | 9019 | 6.1 |
| 7 | Japan | 7734 | 7.2 |
| 8 | Saudi Arabia | 6923 | 17.4 |
| 9 | UK | 6205 | 4.0 |
| 10 | Russia | 5888 | 10.2 |
| | World | 201,818 | - |

The relevance of China is very clear. Iran and Saudi Arabia appear in a relevant place, with a high share of nano-articles to total articles.

*1.2. International Standards for Risk and Safety in Nanotechnology*

The standard ISO 45001:2018—Occupational Health and Safety Management Systems presents very useful information on how to manage this topic in companies. It is aligned with ISO 9001:2015 and other management systems. Concerning nanotechnologies, the ISO has been developing a large set of standards, especially within the ISO Technical Committee 229, created in 2005. Up until now (December 2021), a total of 97 standard documents have been published, and 27 are under development [4].

Working group 3 of the ISO/TC229 deals specifically with Health, Safety, and Environmental Aspects of Nanotechnologies. The following five documents are especially relevant:

- ISO/TR 13121:2011. Nanotechnologies—Nanomaterial risk evaluation. ISO/TR 13121:2011 describes a process for identifying, evaluating, addressing, making decisions about, and communicating the potential risks of developing and using manufactured nanomaterials in order to protect the health and safety of the public, consumers, workers, and the environment. ISO/TR 13121:2011 offers guidance on the information needed to make sound risk evaluations and risk management decisions, as well as how to manage in the face of incomplete or uncertain information by using reasonable assumptions and appropriate risk management practices. Further, ISO/TR 13121:2011 includes methods to update assumptions, decisions, and practices as new information becomes available and on how to communicate information and decisions to stakeholders. ISO/TR 13121:2011 suggests methods that organizations can use to be transparent and accountable in how they manage nanomaterials. It describes a process of organizing, documenting, and communicating what information organizations have about nanomaterials [10,11].
- ISO/TS 12901-1:2012. Nanotechnologies—Occupational risk management applied to engineered nanomaterials—Part 1: Principles and approaches. ISO/TS 12901:2012 provides guidance on occupational health and safety measures relating to engineered nanomaterials, including the use of engineering controls and appropriate personal protective equipment, guidance on dealing with spills and accidental releases, and guidance on the appropriate handling of these materials during disposal. ISO/TS 12901-1:2012 is intended for use by competent personnel, such as health and safety managers, production managers, environmental managers, industrial/occupational hygienists, and others with responsibility for the safe operation of facilities engaged in the production, handling, processing, and disposal of engineered nanomaterials. ISO/TS 12901-1:2012 is applicable to engineered materials that consist of nano-objects such as nanoparticles, nanofibres, nanotubes, and nanowires, as well as aggregates and agglomerates of these materials (NOAA) [10,12].
- ISO/TR 13329:2012—Nanomaterials—Preparation of the material safety data sheet (MSDS). This document provides guidance on the development of content for, and consistency in, the communication of information on safety, health, and environmental matters in safety data sheets (SDS) for substances classified as manufactured nanomaterials and for chemical products containing manufactured nanomaterials. It provides supplemental guidance to ISO 11014:2009 (Safety data sheets for chemical products) on the preparation of SDSs generally, addressing the preparation of an SDS for both manufactured nanomaterials with materials and mixtures containing manufactured nanomaterials [4,13].
- ISO/TS 12901-2:2014. Nanotechnologies—Occupational risk management applied to engineered nanomaterials—Part 2: Use of the control banding approach. ISO/TS 12901-2:2014 describes the use of a control banding approach for controlling the risks associated with occupational exposures to nano-objects and their aggregates and agglomerates greater than 100 nm (NOAA), even if knowledge regarding their toxicity and quantitative exposure estimations is limited or lacking. The ultimate purpose of control banding is

to control exposure in order to prevent any possible adverse effects on workers' health. The control banding tool described here is specifically designed for inhalation control. Some guidance for skin and eye protection is given in ISO/TS 12901 1. ISO/TS 12901-2:2014 is focused on intentionally produced nano-objects, such as nanoparticles, nanopowders, nano fibres, nanotubes, and nanowires, as well as on aggregates and agglomerates of the same. As used in ISO/TS 12901-2:2014, the term NOAA applies to such components, whether in their original form or incorporated in materials or preparations from which they could be released during their lifecycle. ISO/TS 12901-2:2014 is intended to help businesses and others, including research organizations, engaged in the manufacturing, processing, or handling of NOAA, by providing an easy-to-understand, pragmatic approach for the control of occupational exposures [10,14].

- ISO/TR 12885:2018. Nanotechnologies—Health and safety practices in occupational settings. This document describes the health and safety practices in occupational settings relevant to nanotechnologies. This document focuses on the occupational manufacturing and use of manufactured nano-objects and their aggregates and agglomerates greater than 100 nm (NOAAs). It does not address health and safety issues or practices associated with NOAAs generated by natural processes, hot processes, and other standard operations which unintentionally generate NOAAs or potential consumer exposures or uses, although some of the information in this document can be relevant to those areas [10,15].

At the European level, the Technical Committee CEN TC 352 was created in 2006. Under AFNOR/UNMZ (France/Czech Republic) secretariats, the CEN/TC352 is engaged in standardization in the field of nanotechnologies. This includes the development of a set of standards addressing the following aspects of nanotechnologies: classification, terminology, and nomenclature; metrology and instrumentation, including specifications for reference materials; test methodologies; modelling and simulation; science-based health, safety, and environmental practices; nanotechnology products and processes [4].

Under CEN/TC 352 coordination, several CEN Technical Committees are involved in the execution of Mandate M/461 from the European Commission; several European standards have already been published under this Mandate, and several others are in preparation. Many of them have a relation with occupational health and safety.

At present (December 2021), 51 European standards have been published (32 out of these are EN/ISO documents, in conjunction with ISO/TC229), and six are under preparation. The updated list is available online: https://standards.cen.eu/ (accessed on 15 December 2021).

It is also worth mentioning the standards developed within the European Technical Committee CEN/TC 137—Assessment of workplace exposure to chemical and biological agents. The following eight standard documents are in the area of nanotechnology:

- EN ISO 28439:2011—Workplace atmospheres—Characterization of ultrafine aerosols/ nanoaerosols—Determination of the size distribution and number concentration using differential electrical mobility analyzing systems (ISO 28439:2011)
- EN 17058:2018—Workplace exposure—Assessment of exposure by inhalation of nano-objects and their aggregates and agglomerates
- EN 16966:2018—Workplace exposure—Measurement of exposure by inhalation of nano-objects and their aggregates and agglomerates—Metrics to be used such as number concentration, surface area concentration, and mass concentration
- EN 16897:2017—Workplace exposure—Characterization of ultrafine aerosols/nanoaero sols—Determination of the number concentration using condensation particle counters
- CEN ISO/TS 21623:2018—Workplace exposure—Assessment of dermal exposure to nano-objects and their aggregates and agglomerates (NOAA) (ISO/TS 21623:2017)

In 2006, the OECD (Organization for Economic Co-operation and Development) established the Working Party on Manufactured Nanomaterials (WPMN) as a subsidiary body of the OECD Chemicals Committee. This program concentrates on human health and environmental safety implications of manufactured nanomaterials. Since then, OECD has

published more than 100 guidance documents under the series of Safety of Manufactured Nanomaterials [4]. The SOP (Standard Operating Procedures) developed by OECD have been supported at the international level and, therefore, can be used for regulatory purposes.

The full list of all the freely downloadable documents can be consulted at http://www.oecd.org/env/ehs/nanosafety/publications-series-safety-manufactured-nanomaterials.htm (accessed on 15 December 2021).

Note that three of the documents published by the OECD in November 2021 are very relevant: "Evaluation of Tools and Models for Assessing Occupational and Consumer Exposure to Manufactured Nanomaterials" (in 3 parts).

It is important to emphasize the so-called "Malta Initiative" (which arose during the Maltese EU Council Presidency in 2017), involving 18 European countries, in which several Directorate-Generals of the European Commission, the European Chemicals Agency (ECHA), authorities, research institutions, NGOs, universities, and industry work joined together on a voluntary and self-organized basis. The aim of this initiative is to make legislation enforceable, in particular in the chemicals sector. For this purpose, it is necessary to ensure that the essential test, measurement, and verification procedures are available. Currently, the work is focused on amending the OECD Test Guidelines in the area of nanomaterials to ensure that a nanomaterial-adapted REACH Regulation will become enforceable [4].

## 2. Control Banding Approach in Occupational Risk Management Applied to Engineered Nanomaterials

Nanomaterials constitute a new generation of toxic chemicals. As particle size decreases, in many nanomaterials, the production of free radicals increases, as does toxicity [16]. The number of commercial products and the number of workers potentially exposed to engineered nano-materials is growing, as is the need to evaluate and manage the potential health risks [17]. The control banding approach for occupational risk management applied to engineered nanomaterials, according to ISO/TS 12901-2:2014, is a pragmatic approach useful for the control of workplace exposure to possibly hazardous agents with unknown or uncertain toxicological properties and for which quantitative exposure estimations are lacking [4].

The Control Banding process, according to ISO/TS 12901-2:2014, includes the following elements:

- Information gathering;
- Assignment of the nano-objects to a hazard band (on the basis of a comprehensive evaluation of all available data on each material, taking into account parameters such as toxicity; in vivo biopersistence; and factors influencing the ability of particles to reach the respiratory tract, their ability to deposit in various regions of the respiratory tract, and their ability to elicit biological responses);
- Description of potential exposure characteristics (assigning an exposure scenario at a workplace to an exposure band, taking into account the physical form and amount of the nano-object, dust generation potential of processes, and actual exposure measurement data);
- Definition of recommended work environments and handling practices (control banding);
- Evaluation of the control strategy or risk banding.

This standard has been applied in several studies, for instance, in an interesting study that evaluated workers' exposure to nano-objects in R&D laboratories by means of the control banding technique [18].

The authors also applied this methodology in a case study in a textile finishing company involving two chemical finishes containing nanomaterials: a mosquito repellent and antibacterial finish. The risk analysis mainly concerned four workers involved either in the preparation of the finishing baths or on the conducting of the stenter frame. Hazard bands and exposure bands were evaluated, according to ISO/TS 12901-2:2014. Following the application of this standard, the control bands corresponding to the exposure of

the different workers to the nanomaterials were established. As a result, the following measures to mitigate risks were envisaged: appropriate ventilation and use of adequate personal protective equipment. Hazards related to one of the chemicals were higher and also required the use of a closed booth and a smoke extractor [6].

There is now a specific REACH registration system for nanomaterials, which came into force from January 2020 (Commission Regulation (EU) 2018/1881 of 3 December 2018), so it is recommended that the suppliers of chemicals which incorporate nanomaterials include more information on the hazards and measures for risk mitigation in the safety data sheets, based, for instance, on the recommendations presented in ISO/TR 13329. This information is essential for implementing the control banding approach [4].

The new Commission Regulation (EU) 2020/878 of 18 June 2020, amending Annex II to REACH, presents more detailed requirements to be included in the safety data sheets of chemicals that contain nanoforms. This Regulation applies from 1 January 2021, with a transitional period until 31 December 2022 [4].

### 3. Conclusions

This study has presented an overview of the standard documents related to Safety and Risk relevant for Nanotechnologies. The most relevant activities at the international level are related to ISO/TC229. Five standard documents developed within the ISO have been highlighted, and a case study applying one of them has been presented.

At European level, it was suggested to follow the activities of the European Agency for Safety and Health at Work (EU-OSHA). The website of EU-OSHA is the best place to follow all the relevant EU legislation, with links to legislation at the national level, supported by standards.

The increasing concerns related to the health and safety of nanomaterials are leading to the emergence of standards. It is essential that all stakeholders keep aware of all the updated legal requirements and standard documents, considering these not only as limitations, but also as opportunities for improvement.

**Author Contributions:** Both authors have contributed together to all the sections. All authors have read and agreed to the published version of the manuscript.

**Funding:** This research received no external funding.

**Institutional Review Board Statement:** Not applicable.

**Informed Consent Statement:** Not applicable.

**Data Availability Statement:** Not applicable.

**Acknowledgments:** The authors acknowledge support given by the textile company.

**Conflicts of Interest:** The authors declare no conflict of interest.

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
