# Peer review of "Overview of Standards Related to the Occupational Risk and Safety of Nanotechnologies"

_standards, doi:10.3390/standards2010007_

Round 1

Reviewer 1 Report

Dear Author, i would recommend to take into account CEN TC137 production, as several standards could be relevant to this paper. I would also recommend furthering developing limitations of each of the standards developed, notably for the case proposed case study. I would finally recommend furthering developing the case study in order to bring a real add-on value in this paper. 

Author Response

“Dear Author, I would recommend to take into account CEN TC137 production, as several standards could be relevant to this paper. I would also recommend furthering developing limitations of each of the standards developed, notably for the case proposed case study. I would finally recommend furthering developing the case study in order to bring a real add-on value in this paper.

Answer:

Thank you for your comment. Please note that in light of your comments, in the revised version of the paper, we have proceeded to accommodate your comments.

All these changes are shown in the article in blue color.

We have included reference to 8 standards related to nanotechnology published with CEN/TC 137.

Concerning the case study, major findings have been published elsewhere, see for instance ref. 6. But anyhow we have presented more information about the application of the control banding technique, according to ISO/TS 12901-2:2014.

The revised manuscript has been uploaded. We hope you find our revision acceptable and that the paper can now be Approved for publication.

Best regards,

Reviewer 2 Report

Dear Authors,

The review entitled "Overview of Standards Related to Occupational Risk and Safety of Nanotechnologies" is relevant and potentially interesting for the international audience of the journal "Standards", after major changes.

The authors did not ignore the fundamental work done by the OECD, in particular, the OECD Programme on Manufactured Nanomaterials and more recently the "Malta Initiative". However, the emphasis on the ISO standards rather than the OECD standards is not well understood. This is a cornerstone program across the OECD, where EU or Member States funded projects (e.g., NANOMET; NANOHARMONY) have focused on the publication of several freely available data, test guidelines, and guidance documents to assess nanomaterials in a holistic way: https://www.oecd.org/science/nanosafety/publications-series-safety-manufactured-nanomaterials.htm

Therefore, authors are invited to explain why this markable joint effort across the OECD countries was underestimated and limited to just 33 lines of your review. If the intent is to classify the manuscript as "review", the Authors need to explore the OECD documents focused on the framework of NMs occupational and humans exposure risk management, and critically discuss the positive and negatives aspects, the gaps, and the critical differences among the ISO, OECD, and other standards.

The current document is too simple, short, and scientifically narrow having just 14 (!) references, the large majority referring to standards or websites. Furthermore, 3 out of 4 cited references are auto-citations, including not-so-accessible chapter books. Authors are invited to bring other studies where such standards were implemented and critically discuss them. 

Figures 1 and 2 are copy-paste from the website. Have the Authors requested copyright permission to publish such figures? Please, reconsider making your own graphs, reporting that they are based on the data provided by StatNano. 

On the other hand, the paragraph related to the number of publications focused on Nanotechnology (lines 76-78) and respective Table 1 highlight the United Kingdom and Portugal scientific production. Why these two countries? Table 1 would be more accurate if listed the Top 40 or 50 countries if the objective is to show the position of Portugal. Otherwise, it will look quite biased and regional. Authors are also invited to enrich this section by adding much more information, namely source (where the website collected such data?), sub-fields, or main subjects (you can dig a bit in the ISI Web of Knowledge or Scopus). 

Finally, the Authors jump from chapter 2 to chapter 5 (conclusions). Is there any information missing?

All the best.

Author Response

03.02.2022

Reviewer 2#

This paper is the amended manuscript ID standards-1532626 version R1 of the manuscript Overview of Standards Related to Occupational Risk and Safety of Nanotechnologies”.

Below are the answers to your comments.

“Dear Authors,

The review entitled "Overview of Standards Related to Occupational Risk and Safety of Nanotechnologies" is relevant and potentially interesting for the international audience of the journal "Standards", after major changes.”

Thank you for your comment. Please note that in light of your comments, in the revised version of the paper, we have proceeded to accommodate your comments.

All these changes are shown in the article in blue color.

“The authors did not ignore the fundamental work done by the OECD, in particular, the OECD Programme on Manufactured Nanomaterials and more recently the "Malta Initiative". However, the emphasis on the ISO standards rather than the OECD standards is not well understood. This is a cornerstone program across the OECD, where EU or Member States funded projects (e.g., NANOMET; NANOHARMONY) have focused on the publication of several freely available data, test guidelines, and guidance documents to assess nanomaterials in a holistic way: https://www.oecd.org/science/nanosafety/publications-series-safety-manufactured-nanomaterials.htm

Therefore, authors are invited to explain why this markable joint effort across the OECD countries was underestimated and limited to just 33 lines of your review. If the intent is to classify the manuscript as "review", the Authors need to explore the OECD documents focused on the framework of NMs occupational and humans exposure risk management, and critically discuss the positive and negatives aspects, the gaps, and the critical differences among the ISO, OECD, and other standards.”

The title of the paper is an overview of standards, so authors have focused the paper on ISO and EN standards. OECD documents are not really standards but reports or guidance documents. Nevertheless, the text concerning OECD has been revised.

“The current document is too simple, short, and scientifically narrow having just 14 (!) references, the large majority referring to standards or websites. Furthermore, 3 out of 4 cited references are auto-citations, including not-so-accessible chapter books. Authors are invited to bring other studies where such standards were implemented and critically discuss them.”

The paper has been prepared on the experience of the authors in the area. Note that Dr. Almeida is President of the Portuguese Technical Committee for Standardization of Nanotechnologies and follows the CEN/TC352 and ISO/TC229 Technical Committees. Most of the information in the paper is related to this work, that is why there is a limited number of references. Anyway, new four references have been added, all of scientific papers, including one of a paper published in 2022. All the reference numbers have been updated.

“Figures 1 and 2 are copy-paste from the website. Have the Authors requested copyright permission to publish such figures? Please, reconsider making your own graphs, reporting that they are based on the data provided by StatNano. “

Figures 1 and 2 have been deleted. Figure 1 has been replaced by Table 1. Old Table 1 has been renumbered as Table 2.

“On the other hand, the paragraph related to the number of publications focused on Nanotechnology (lines 76-78) and respective Table 1 highlight the United Kingdom and Portugal scientific production. Why these two countries? Table 1 would be more accurate if listed the Top 40 or 50 countries if the objective is to show the position of Portugal. Otherwise, it will look quite biased and regional. Authors are also invited to enrich this section by adding much more information, namely source (where the website collected such data?), sub-fields, or main subjects (you can dig a bit in the ISI Web of Knowledge or Scopus). “

Table 1 (now Table 2) is a revised extract of the information supplied by Statnano, with all data of 2021, avoiding the biased and regional information concerning the highlighting of Portugal. The percentage of nano-articles respective to total has also been added.

Finally, the Authors jump from chapter 2 to chapter 5 (conclusions). Is there any information missing?”

Sorry for this mistake. Section 5 (conclusions) has been renumbered as section 3 (conclusions).

We express our special thanks to you for careful reading of the paper, and for providing

valuable comments and suggestions which have helped to improve both the content and the presentation.

The revised manuscript has been uploaded. We hope you find our revision acceptable and that the paper can now be Approved for publication.

Best regards,

Round 2

Reviewer 1 Report

Dear Author, thank you for the improvments made.

I have any minors suggestions :

line 196 : OECD is developing technical guidlines or guidance documents, describing a SOP rather than the way to use an instrument. These documents have been agreed at international level and therefore can be used for regulatory purpose (for example Reach dossier request data produced using theses SOPs). Normaly, in a ideal world, standards (ISO, CEN) are supporting guidlines... I would then suggest to adjust in accordance

Author Response

 Reviewer 1#

This paper is the amended manuscript ID standards-1532626 version R2 of the manuscript Overview of Standards Related to Occupational Risk and Safety of Nanotechnologies”.

Below are the answers to your comments.

“Dear Author, thank you for the improvements made.

I have any minor suggestions:

line 196 : OECD is developing technical guidelines or guidance documents, describing a SOP rather than the way to use an instrument. These documents have been agreed at international level and therefore can be used for regulatory purpose (for example Reach dossier request data produced using theses SOPs). Normally, in a ideal world, standards (ISO, CEN) are supporting guidelines... I would then suggest to adjust in accordance.”

Answer:

Thank you for your comment. Please note that in light of your comments, in the revised version of the paper, we have proceeded to accommodate your comments. The paragraph concerning OECD has been adapted accordingly (see lines 194-197 highlighted in blue).

The revised manuscript has been uploaded. We hope you find our revision acceptable and that the paper can now be Approved for publication.

Best regards,

Reviewer 2 Report

Dear Authors,

Thanks for your revision, which benefitted the final quality of the manuscript. 

The authors should reconsider changing the title since does not accurately reflect the content of the manuscript. It should be less broad. "Standards related to occupational risk and safety of nanotechnologies: implementation of the ISO/TS 12901-2:2014 as a case study."

References are not harmonized in terms of style. 

Author Response

Reviewer 2#

This paper is the amended manuscript ID standards-1532626 version R2 of the manuscript Overview of Standards Related to Occupational Risk and Safety of Nanotechnologies”.

Below are the answers to your comments.

“Dear Authors,

Thanks for your revision, which benefitted the final quality of the manuscript.

The authors should reconsider changing the title since does not accurately reflect the content of the manuscript. It should be less broad. "Standards related to occupational risk and safety of nanotechnologies: implementation of the ISO/TS 12901-2:2014 as a case study."

References are not harmonized in terms of style.

Answer:

Thank you for your comments.

In fact, the objective of the paper is to present an overview of the standards related to occupational risk and safety of nanotechnologies. Details of the case study have been published elsewhere (see for instance reference 6). In the present paper we have only included a short reference to this case study in a paragraph (lines 244 to 253), so the inclusion of this reference in the title would be misleading to the readers. Mention to the case study has been removed from the abstract, as it could also be misleading, Changes in the abstract are highlighted in blue.

The style of the references has been harmonized.

We express our special thanks to you for careful reading of the paper, and for providing

valuable comments and suggestions which have helped to improve both the content and the presentation.

The revised manuscript has been uploaded. We hope you find our revision acceptable and that the paper can now be Approved for publication.

Best regards,
